# Adaptive Domain Learning for Cross-domain Image Denoising

**Zian Qian**[1]    **Chenyang Qi**[1]    **Ka Lung Law**[2]    **Hao Fu**[2]    **Chenyang Lei**[3†]    **Qifeng Chen**[1†]

[1]HKUST    [2]SenseTime    [3]CAIR, HKISI-CAS

{zqianaa, cqiaa, cqf}@ust.hk
{luojialong, fuhao1}@sensetime.com
leichenyang7@gmail.com

## Abstract

Different camera sensors have different noise patterns, and thus an image denoising model trained on one sensor often does not generalize well to a different sensor. One plausible solution is to collect a large dataset for each sensor for training or fine-tuning, which is inevitably time-consuming. To address this cross-domain challenge, we present a novel adaptive domain learning (ADL) scheme for cross-domain RAW image denoising by utilizing existing data from different sensors (source domain) plus a small amount of data from the new sensor (target domain). The ADL training scheme automatically removes the data in the source domain that are harmful to fine-tuning a model for the target domain (some data are harmful as adding them during training lowers the performance due to domain gaps). Also, we introduce a modulation module to adopt sensor-specific information (sensor type and ISO) to understand input data for image denoising. We conduct extensive experiments on public datasets with various smartphone and DSLR cameras, which show our proposed model outperforms prior work on cross-domain image denoising, given a small amount of image data from the target domain sensor.

## 1 Introduction

Noise generated by electronic sensors in a RAW image is inevitable. Over the past few years, learning-based methods have made significant progress in RAW image denoising [5, 22, 25, 42]. However, building a large-scale real-world dataset with noise-clean pairs for training a denoising model is time-consuming and labor-intensive. It is hard to collect ground truth that is noise-free and has no misalignment with the input noisy data. Moreover, due to the different noise distributions of different sensors (such as read noise and shot noise), the collected data from a particular sensor usually cannot be used to train the denoising model of other sensors, which causes a waste of resources. Therefore, it is important to develop a method to solve this problem.

Existing solutions to data scarcity in RAW image denoising can be divided into two categories, noise calibration [42, 47, 27] and self-supervise denoising [22, 18, 25, 41, 20]. Noise synthesis and calibration methods first build a noise model, optimize for noise parameters according to a particular camera, and then synthesize training pairs from the noise model to train a network. Self-supervised denoising is designed based on the blindspots schemes. When the input noisy image masks out some pixels and forms a similar but different image from the input, the network learns to denoise instead of identity mapping. Therefore, the network can learn to denoise without pairwise noise-clean data.

---

† Corresponding author.

38th Conference on Neural Information Processing Systems (NeurIPS 2024).

While the noise synthesis and calibration methods are top-performing ones for RAW data denoising and self-supervised denoising does not need to collect pairwise data, both of them have their practical limitations. First, noise synthesis and calibration methods are not able to obtain the exact noise model of the real noise. For example, fixed pattern noise such as dark signal non-uniformity (DSNU) and Photo-response non-uniformity (PRNU) are not included in the model. As a result, some of the sampled noise training pairs might be harmful to the training of the denoising model (i.e., decrease in performance). Second, building a calibration model still needs to collect data under particular circumstances. Third, these models can only be used to synthesize training data for specific sensors, which leads to a waste of resources. On the other hand, self-supervised denoising is designed under some unverified assumptions of noise distribution. First, the noise distribution has zero means. Second, the noise in different pixels is independent of each other. These assumptions do not match the noise in the real world, especially when the noise distribution is complicated. Therefore, self-supervised denoising does not achieve state-of-the-art denoising performance.

Different from prior work, we solve this problem by proposing a cross-domain RAW image denoising method, adaptive domain learning (ADL). Our method can utilize existing RAW image denoising datasets from various sensors (source domains) combined with very little data from a new sensor (target domain) together to train a denoising model for that new sensor. Some data in a source domain may be harmful to fine-tuning a model due to the large domain gap: for instance, synthetic data may be harmful to training a model for real-world applications if the synthetic data imposes unrealistic and unreasonable assumptions. In such cases, our method dynamically evaluates whether a data sample from a source domain is beneficial or harmful by evaluating the performance on a small validation set of the target domain, before and after fine-tuning the model on this data sample. If the performance improves after fine-tuning, we can use this data sample for training; otherwise, we should ignore it. As for the network architecture, we design a modulation network that takes sensor-dependent information as input (sensor types and ISO), which aligns the features from different sensors into the same space and ensembles useful common knowledge for denoising.

To evaluate our proposed model with ADL, we compare our model against prior methods on diverse real-world public datasets [2, 42, 5] captured by both smartphone and DLSR cameras. The results demonstrate that our method outperforms the prior work and shows consistent state-of-the-art performance with ADL on RAW data denoising, given a small amount of data in the target domain. We also demonstrate that our ADL can be applied to fine-tuning existing noise calibration models with cross-domain data to further improve its performance.

The contributions of this work can be summarized as follows.

- We propose a novel adaptive domain learning (ADL) strategy that can train a model with little data from a new sensor (target domain), by automatically leveraging useful information and removing useless data from existing RAW denoising data from other sensors (source domains).

- A customized modulation strategy is applied to provide sensor-specific information, which helps our network adapt to different sensors and noise distributions.

- Our model outperforms prior methods in cross-domain image denoising in the target domain with little data.

## 2 Related Work

### 2.1 Raw Data Denoising

In recent years, methods based on RAW data denoising draw a lot of attention [29, 5, 42, 47, 4, 44, 25, 1, 48, 27]. SID [5] shows that RAW image denoising can perform well with a naive U-net architecture. Besides, they find difficulties in collecting large-scale datasets. Aware that collecting datasets is the research bottleneck, many approaches attempt to synthesize more realistic data. UIP [4] and CycleISP [44] attempt to inverse the image signal processing pipeline and synthesize noise in RAW space to train a RAW denoising framework. However, the generated pseudo-RAW data still has great differences compared to real RAW data. Jin et al. [16] utilize different noise distribution parameters to form a simulation camera to train a network. However, they still need to build a noise model to synthesize data. Another kind of approach is the noise calibration method [42, 47, 27, 45].

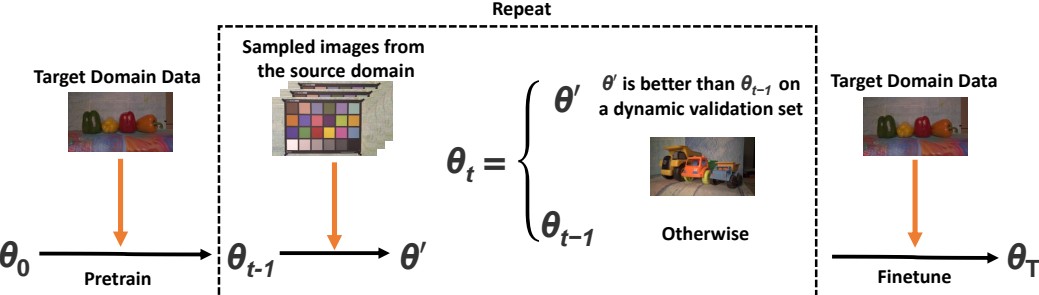

Figure 1: **The overall pipeline of our adaptive domain learning (ADL) algorithm.** The network parameter $\theta_0$ is first initialized, then the small target domain training set will be used to train a model with parameter $\theta$. In the source domain adaptive learning stage, in iteration $t$, data from the source domain will be used to update the network parameter from $\theta_{t-1}$ to $\theta'$. Then a dynamic validation set will judge whether the data is useful. If so, set $\theta_t = \theta'$ and repeat the process. If not, retrieve the network parameter from $\theta'$ to $\theta_{t-1}$. Finally, the target domain data will be used to fine-tune $\theta'$ to $\theta_T$
.

Wei et al. [42] and Zhang et al. [47] analyze the noise source from the electronic pipeline of the DSLR cameras and build corresponding noise model to synthesize data. However, the domain gap still exists between the synthetic noise and the real noise. Besides, the noise calibration model is usually designed for a specific sensor, and hence no generalization ability and is not reusable. Our ADL can overcome the domain gap between real and synthetic data by removing harmful data. We can also utilize our ADL algorithm to fine-tune the existing noise calibration method to further improve its performance. Lehtinen et al. [22] find that pairs of two low-quality images with very similar content are enough to train a denoise model, forming the self-supervised denoising field. As the modified version of Noise2noise, methods that can learn to denoise with a single image are developed [18, 21, 41, 12, 20, 31, 14, 45]. They design a blind-spot network to force the model to learn the mapping from noisy to noise-free. In such cases, no paired data is needed for training. However, their method has some assumptions about noise distribution, which is usually not the case in the real world. Due to the above constraints, these methods usually cannot reach state-of-the-art performance compared to other noise-to-clean supervised methods. Our method can solve the problem of data collection while keeping state-of-the-art performance.

## 2.2 Meta-transfer learning in low-level vision

The gap between different domains (synthetic and real, daylight and night, etc.) is a great challenge in the field of low-level vision. To solve the problem, a set of approaches based on meta-transfer learning is proposed [32, 35, 17, 6, 43, 39, 40, 34, 15, 8]. Park et al. [32] and Soh et al. [35] utilize MAML algorithm [9] in super-resolution to obtain a model with better initialization implicitly from the source domain, then fine-tune it to fit the target domain better. Kim et al. [17] transferred the useful features from the synthesis noise model to the real-world noise model to overcome the domain gap problem between real-world noise and synthetic noise with an adaptive instance normalization layer [37, 13, 24, 33] to help the synthetic noise better adapt to real-world noise. For the meta-transfer learning method, it is very hard to tell whether the implicitly learned information is useful or not. Some of the transferred features might be harmful. In contrast, our ADL can learn common knowledge and remove harmful information explicitly.

## 3 Method

In this section, we introduce the three steps in our adaptive domain learning pipeline: target domain pretraining, source domain adaptive learning, and target domain fine-tuning. The overall pipeline is illustrated in Figure 1.

### 3.1 Adaptive Domain Learning Algorithm

#### 3.1.1 Target Domain Pretraining

Given the small training set of the target domain $T^{adp}$, we first pre-train a model for the target domain by minimizing pixel-wise $L_1$ Loss.

Target domain pre-training has benefits in two aspects. First, although the domain gap exists between the source domain and the target domain, denoising is a task that shares similar implicit feature representations. Therefore, pre-training can provide better initialization for the adaptive domain learning stage. Second, there is only very little data from the target domain, and the data from the source domain can be 100 times more than the data in the target domain. Pre-training on the target domain can improve the robustness and ensure the dominant position of the target domain data in the whole training process to prevent our model from overfitting to the source domain.

#### 3.1.2 Source Domain Adaptive Learning

However, due to the domain gap between the source domain and the target domain, not all the data from the source domain contribute to the training of the target domain, some data might be harmful and will lead to performance reduction. Therefore, we proposed adaptive domain learning (ADL), to eliminate harmful data and make use of the one that has contributions to our model.

In each iteration $t$, we sample a batch of training data $S'$ from some source domain $S(i)$, and adapted our pretrained parameter $\theta_{t-1}$ to $\theta'$ by

$$\theta' \leftarrow \theta_{t-1} - \alpha \nabla_{\theta_{t-1}} \mathcal{L}(S'), \tag{1}$$

where $\alpha$ is the learning rate and $\mathcal{L}(S')$ is the $L_1$ loss defined on $S'$.

**Dynamic validation set** To tell whether the data batch $S'$ has contributions to our model, we evaluate the updated parameter $\theta'$ on a target domain validation set $T^{val}$ that is sampled from the target domain dataset $T^{adp}$. The selection of the validation set in each iteration $t$ is crucial to the performance of our method. Fixed validation set selection for each iteration may make the training stuck in the local minima and easily overfit to the validation set. To avoid these problems from happening, in each iteration $t$, we randomly sampled a dynamic validation set $V'$ of size $k$ from the target domain dataset $T^{adp}$ to let our model explore the feature space in a stochastic way. On the other hand, the rest of the dataset from $T^{adp}$, denoted as $T^{Train}$, will combined with $S'$ to provide the correct direction for the training process. At the beginning of the training, $k$ is set to 20% of the size of $T^{adp}$ and increases during the training process. At the end of the training, 50% of $T^{adp}$ will be used.

Moreover, inspired by [16], when the size of $T^{adp}$ is extremely small, i.e., smaller than 10, we intentionally select the data that has very diverse system gain from $T^{adp}$ to form $V'$ in each iteration to avoid the over-fitting problem.

**Dynamic average PSNR** We evaluate whether the data batch $S'$ is useful or not by comparing the PSNR of the result of the updated network parameter $\theta'$ to the PSNR of the result of previous iteration $\theta_t$ on the sampled validation set $V'$. However, hard criteria based on PSNR usually make the training procedure unstable under the setting of the dynamic validation set. When the size of the dynamic validation set $V'$ is small, the variance of PSNR is large and thus is not that reliable. Some useful data might be removed accidentally. In such cases, we want our model to take a data batch $S'$ as useful data if $S'$ has a trend to improve the performance of our model. We design soft criteria based on PSNR by maintaining a priority queue $Q^{eval}$ of max size $M$ that stores the value of highest PSNR value in the history during the training process. $Q^{eval}$ is ranked by the value of PSNR in ascending order. We denote the PSNR on our dynamic validation set $V'$ of model $\theta'$ at iteration $t$ in our training process as $Eval(V', \theta_t)$. At the beginning of the adaptive domain learning stage, we push $Eval(V', \theta_{t=0})$ into $Q^{eval}$. During the training, if the PSNR of the updated parameter $\theta'$ on the dynamic validation set $V'$, $Eval(V', \theta')$, is higher than the average PSNR in $Q^{eval}$, we keep the updated parameter $\theta'$ and push $Eval(V', \theta')$ into $Q^{eval}$ and pop out the first element in $Q^{eval}$ if it is full. Else, we retrieve the network parameter from $\theta'$ to $\theta_{t-1}$. This process can be characterized as

$$\theta_t = \begin{cases} \theta', & Eval(V', \theta') > \dfrac{1}{m} \sum_{i=1}^{m} Q_i^{eval} \\ \theta_{t-1}, & otherwise \end{cases} \tag{2}$$

---

**Algorithm 1** Adaptive domain learning (ADL)

---

**Require:** $S_1, \ldots, S_n$: training sets of $n$ source domains
**Require:** $T^{adp}$: The small target domain dataset

**Require:** $Q^{eval}$: priority queue with max length $M$ that stores the PSNR.
1:  Initialize a model of $\theta_0$ by pretraining on $T^{adp}$
2:  **for** $t \leftarrow 1$ to $T$ **do**
3:      Randomly sample images $S'$ from some domain $S_i$
4:      Randomly sample images $V'$ from $T^{adp}$, the rest part $T^{train} = T^{adp} - V'$
5:      Merge $S'$ and $T^{train}$ by $S' = T^{train} + S'$
6:      $\theta' \leftarrow \theta_{t-1} - \alpha \nabla_{\theta_{t-1}} \mathcal{L}(S')$
7:      **if** $Eval(V', \theta') > \frac{1}{m} \sum\limits_{i=1}^{m} Q_i^{eval}$ **then**
8:          $\theta_t = \theta'$
9:          **if** $Q.size() == M$ **then**
10:             $Q^{eval}.pop()$
11:         $Q^{eval}.push(Eval(V', \theta'))$
12:     **else**
13:         $\theta_t = \theta_{t-1}$
14: Fine-tune the model of $\theta_T$ on $T^{adp}$

---

In the final stage, we fine-tune the network parameter $\theta'$ obtained in the previous stage using the target domain training set and update the network parameter to $\theta_T$. The detail of our adaptive domain learning algorithm is illustrated in Algorithm 1

## 3.2   Channel-wise Modulation Network

To let our network better utilize the information from sensors that have different noise distributions, we need to adjust the feature space of different inputs. For a RAW data $D$ captured by a CMOS sensor, we can model its noise by:

$$N = I + KN_{dep} + N_{indep}, \tag{3}$$

where $K$ is the system gain, $N_{dep}$ is the signal dependent noise and $N_{indep}$ is the signal independent noise. Based on this modeling, we propose a channel-wise modulation network to adjust the feature space by embedding two easy-to-access parameters, the sensor type and the ISO in our network. ISO is proportional to system gain $K$, while the sensor type can help the network know how to utilize the ISO to learn the signal-dependent noise $N_{dep}$ and recognize the signal-independent noise $N_{indep}$.

Given the one-hot encoding of the sensor type $p \in R^{1 \times n}$ and the corresponding ISO $s \in R^{1 \times n}$ (duplicate $n$ times in the vector), our channel-wise modulation layer transfers the concatenated metadata $(p, s)$ into a channel-wise scale $\gamma$ and shifts $\beta$ by

$$\gamma = 1 + \tanh(\text{MLP}_\gamma(p, s)), \tag{4}$$
$$\beta = \text{MLP}_\beta(p, s), \tag{5}$$

where $\gamma, \beta \in R^{1 \times C}$, and $\text{MLP}_\beta$ and $\text{MLP}_\gamma$ are two four layer Multi-Layer Perceptrons. Let the feature map of the $i$-th convolution layer be $F_i \in R^{H \times W \times C}$, we embed the sensor-specific data to $F_i$ by a channel-wise linear combination by

$$F_i' = \gamma \times F_i + \beta. \tag{6}$$

Note that the type of the input metadata of our channel-wise modulation strategy is not fixed. The input concatenated vector can be extended as long as more meta information is provided along with the data.

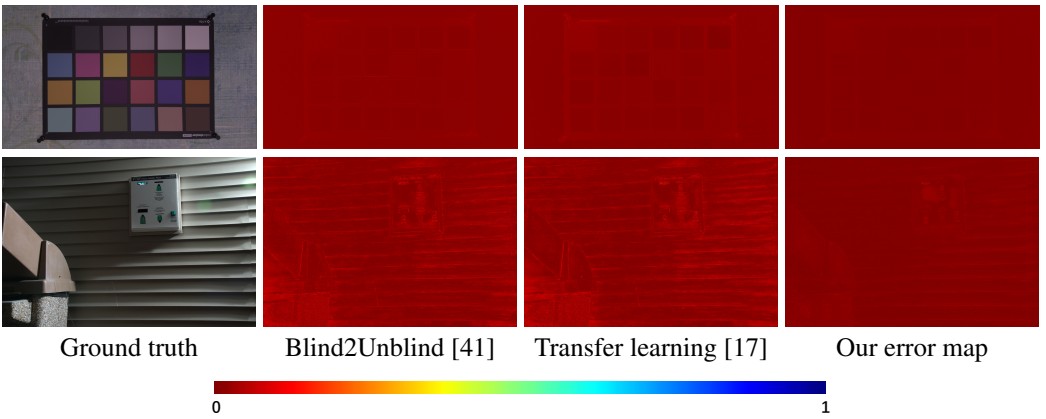

| Ground truth | Blind2Unblind [41] | Transfer learning [17] | Our error map |

Figure 2: **The error map of our method compares against state-of-the-art approaches.** The first row is the result from the SIDD dataset, and the second row is the result from the SID dataset. We can see that our method is able to generate the image with smaller errors and less noise compared to previous work.

## 4    Experiments

### 4.1    Experimental Setup

**Datasets.** We evaluate the performance of our ADL and modulation on the dataset captured by smartphones in normal light conditions (SIDD dataset [2]), and the dataset captured by DLSR cameras in extremely low light conditions (ELD [42] and SID [5] dataset). Compared to RAW data captured by smartphones, RAW data captured by DSLR cameras in extremely low light environments is more difficult to denoise because the noise distribution is more complicated and the noise level is larger. Moreover, the domain gap between the RAW data captured by different DSLR cameras is larger than the domain gap between the RAW data captured by smartphones.

SIDD [2] is a popular RAW denoising dataset that contains 160 pairs of noisy and ground-truth RAW data from 5 different smartphone cameras (G4, GP, IP, N6, S6) of different scenes. ELD Dataset [42] contains RAW data captured by 3 different brands of DSLR cameras (Nikon, Canon, Sony) with different ISO and light factors, while the SID dataset captured RAW data using 2 different brands of DSLR cameras (Sony and Fuji). Note that the ELD dataset and SID dataset [5] are using the same DSLR camera (Sony A7S2). We only use the data captured by this camera from SID dataset to keep the domain gap between each set of domains. Besides, the input RAW data of the ELD and SID datasets are captured in extremely low light environments, while the ground truth is captured in normal light conditions. We follow the training strategy in [5] by multiplying a light factor by the input RAW data to keep the input RAW data and ground-truth RAW data in the same space.

**Baselines and training settings.** To evaluate the performance of our framework, we compare our method against several baselines: the fundamental baselines pre-train and then fine-tune. Self-supervise denoising methods: Blind2unblind [41], ZS-N2N [26], and DIP [38]. Meta transfer learning method MZSR [35], Prabhakar et al. [34] and Kim et al. [17] (denoted as transfer learning). Traditional approach BM3D [7]. Calibration Free Method Led [16]. Note that we try our best to find all possible work that has the same goal as our method for the comprehensive baseline comparisons. Although the approach might be different, we make the experiment as fair as possible with proper settings.

We compare the performance of our method against the above baselines by cross-validation on each sensor of all three datasets. In each experiment, we take one sensor as the target domain to represent the sensor with a very small number of data (around 20 pairs of data). The data from all other sensors will form the source domain, which represents the existing dataset. For the baselines, we only use the data from the target domain for the training of all self-supervised denoising methods, no data from the source domain is used since cross-domain data will reduce the overall performance of these methods. For fine-tuning, we first pre-train the model with the data from the source domain using U-net, then use the data from the target domain for fine-tuning. For DIP [38], the model is trained on

| Method | G4 | GP | IP | N6 | S6 | Avg. |
|---|---|---|---|---|---|---|
| Fine-tuning | 50.17/0.968 | 43.53/0.914 | 52.77/0.977 | 43.86/0.917 | 37.88/0.863 | 45.58/0.928 |
| BM3D [7] | 50.08/0.968 | 42.14/0.909 | 52.39/0.972 | 43.40/0.916 | 35.52/0.855 | 44.71/0.924 |
| DIP [38] | 46.91/0.931 | 39.88/0.896 | 48.81/0.955 | 41.73/0.906 | 35.23/0.855 | 42.51/0.909 |
| ZS-N2N [26] | 48.86/0.941 | 41.54/0.909 | 50.06/0.968 | 41.88/0.910 | 35.07/0.856 | 43.48/0.917 |
| MZSR [35] | 51.84/0.972 | 44.58/0.921 | 53.74/0.982 | 45.07/0.924 | 37.21/0.868 | 46.49/0.933 |
| Transfer learning [17] | 52.28/0.974 | 44.96/0.923 | 53.04/0.982 | 44.77/0.923 | 40.10/0.898 | 47.03/0.940 |
| Blind2Unblind [41] | 51.78/0.970 | 44.91/0.919 | 54.12/0.985 | 46.02/0.928 | 38.85/0.892 | 47.14/0.939 |
| Prabhakar et al. [34] | 51.76/0.972 | 44.68/0.919 | 53.82/0.983 | 44.92/0.922 | 38.67/0.878 | 46.34/0.933 |
| **Ours** | **52.55/0.975** | **45.18/0.923** | **54.37/0.987** | **46.13/0.932** | **40.16/0.901** | **47.68/0.944** |

| Method | Sony | Fuji | Nikon | Canon | Avg. |
|---|---|---|---|---|---|
| Fine-tuning | 35.94/0.857 | 36.37/0.862 | 35.22/0.853 | 35.63/0.855 | 35.79/0.857 |
| BM3D [7] | 35.61/0.856 | 35.88/0.857 | 35.37/0.853 | 35.07/0.852 | 35.48/0.855 |
| DIP [38] | 31.02/0.696 | 29.44/0.611 | 30.71/0.652 | 30.53/0.641 | 30.42/0.650 |
| ZS-N2N [26] | 32.15/0.724 | 30.39/0.632 | 30.46/0.643 | 31.34/0.707 | 31.09/0.677 |
| MZSR [35] | 36.21/0.861 | 36.98/0.866 | 36.14/0.860 | 35.89/0.857 | 36.31/0.861 |
| Transfer learning [17] | 36.92/0.864 | 37.33/0.869 | 36.49/0.862 | 35.77/0.858 | 36.63/0.863 |
| Blind2Unblind [41] | 36.71/0.866 | 36.57/0.866 | 35.88/0.857 | 35.49/0.855 | 36.16/0.861 |
| Prabhakar et al. [34] | 36.12/0.859 | 36.33/0.864 | 35.47/0.854 | 35.72/0.857 | 36.01/0.861 |
| **Ours** | **37.28/0.871** | **37.58/0.872** | **36.74/0.866** | **36.45/0.868** | **37.01/0.868** |

Table 1: **The quantitative PSNR and SSIM results compared to the baselines on the SIDD (G4, GP, IP, N6, and S6), ELD (Nikon, and Canon), and SID (Fuji and Sony) datasets.** "Fine-tuning" means training on source domain data and then fine-tuning on target domain data. The camera name on the top row means that we keep this camera as the target domain and use data from other cameras as the source domain.

each test data. We first split the whole dataset into two parts, the training set, and the test set. The training set $T^{adp}$ is used for the training of the baselines. All training RAW data will first be packed into 4-channel according to the Bayer pattern and then cropped into patches with shape $256 \times 256$. We train our ADL in 300k iterations, using AdamW as the optimizer with a learning rate of $3 \times 10^{-3}$.

## 4.2 Results on Real Data

We quantitatively evaluate our method by comparing the PSNR against other baselines on the smartphone dataset SIDD and DSLR camera dataset ELD and SID. Note that we only report PSNR [11] because there are no other systematic evaluation metrics designed for RAW data. Other popular evaluation metrics like LPIPS [46] is not suitable for RAW data. We also present the SSIM [11] metrics in the supplementary material. The result is illustrated in Table 1. Here, "fine-tuning" denotes the experiment training on the source domain and fine-tuning on the target domain. The camera name on the top row means that we take this sensor as the target domain while keeping the other sensors as the source domain. For example, "G4" in Table 1 means that this experiment takes G4 as the target domain and all data from the other four sensors, GP, IP, N6, and S6 will form the source domain dataset. From the table, we can see that our proposed method has the best performance on both the smartphone dataset and the DSLR dataset. To be specific, our method has $0.71dB$ and $0.39dB$ performance gains on average compared to MZSR [35] and transfer learning [17] baseline. The PSNR values in the table of the ELD and SID dataset are much lower than those in the table of the SIDD dataset because the ELD and SID datasets are captured in extremely low light environments: the noise level is higher, and the scenes are much more complicated than the SIDD dataset. Note that although the AIN module in Kim et al. [17] has a similar function and architecture to our modulation strategy, they can only estimate the noise level, which is very limited when the source domain contains data from many sensors. Our modulation strategy is more extensible and can somehow provide more hyper information(if provided in the dataset) and can let the network know the difference between the sensors.

**Qualitative Result** Since the PSNR of the RAW denoising is relatively high and difficult to tell the difference from the human visual perspective, We evaluate our method qualitatively by comparing the error map against all three baselines. As illustrated in Figure 2, since the RAW data is hard to

| Dataset | Sensor | Zhang et al. | Single | | Multiple | |
|---|---|---|---|---|---|---|
| | | | FT | ADL | FT | ADL |
| SIDD | GP | 45.36 | 45.47 | **45.62** | 45.32 | **45.83** |
| | S6 | 43.17 | **43.45** | 43.44 | 42.66 | **43.69** |
| | IP | 54.93 | 55.24 | **55.37** | 55.11 | **55.68** |
| ELD | Sony | 44.86 | 44.93 | **44.98** | 44.68 | **45.17** |
| | Nikon | 43.21 | 43.26 | **43.34** | 42.96 | **43.54** |

| Sensor | Led [16] | Ours |
|---|---|---|
| Sony | 36.89 | 37.28 |
| Fuji | 36.95 | 37.58 |
| Nikon | 36.26 | 36.74 |
| Canon | 36.17 | 36.45 |

(a) **The PSNR result of applying our ADL to fine-tune the existing noise calibration model.** "FT" means naive fine-tuning, "Single" means only using data from the corresponding sensor to fine-tune, while "Multiple" means using data from all sensors in the corresponding dataset to fine-tune.

(b) **The PSNR result of our ADL comparing to Led on the existing data from various sensors.**

Table 2: The analysis of calibration model and non-calibration model. The SSIM result is included in the supplementary material.

visualize, we transfer the ground truth into the sRGB domain using LibRAW to better demonstrate the color and details. For the restored RAW images, it is hard to use a well-designed ISP (Image Signal Processing) pipeline to obtain visually pleasant sRGB images for different sensors, we only demonstrate the result on RAW space. The error map is calculated in RAW space. We can see that the error between the ground truth and our output noise-free image is much smaller compared to all baseline methods. For more qualitative results, please refer to supplementary material.

## 4.3 Analysis of Calibration-related Methods

**Noise calibration methods** Although the noise calibration method is powerful, the calibrated model still does not include out-of-model noise such as fixed-pattern noise. Thus, fine-tuning using real data can further improve the performance of the model trained by those synthetic data. However, when the real data used for fine-tuning is scarce, the improvement is usually marginal. In such cases, we may want to utilize more data from different domains to further improve the result. In this section, we analyze the performance of applying our ADL to fine-tune the existing noise calibration model with data from multi-domain. We utilize the noise calibration method proposed by Zhang et al. [47]. The result is illustrated in Table 2 (a). Here "Single" means that only the data from that corresponding sensor is used for fine-tuning, while "Multiple" means that the data from all sensors in that dataset is used for fine-tuning. The result shows that when utilizing limited data from a single domain, the improvement of both naive fine-tuning and ADL is marginal. When we use more data from multiple domains, naive fine-tuning cannot learn useful information from various domains and thus leads to a drop in the final performance. However, our ADL can remove the harmful data, and ensemble the useful information from the different domains to help the training.

**Calibration-free methods** Different from the noise calibration method, the calibration-free method Led [16] embed noise distribution from the simulated camera into the pre-train model. Although LED is a calibration-free method, the network also has no prior knowledge of the noise distribution of the target domain data during the pre-train stage. The large intensity distribution gap between the simulation cameras and the test set will lead to low performance. However, our ADL has prior knowledge of the target domain noise intensity distribution throughout the training process, which can gain similar robustness to the calibration method. As illustrated in Table 2 (b), we replace the synthetic data from the well-calibrated simulated camera in Led [16] with the data from the source domain(data from existing sensors), which is the same as our method in the experiments. Our method can outperform Led [16] in this case. This is because the performance of Led [16] highly depends on the prior knowledge of noise distribution learning from the simulated camera. Our method will not use data with a huge gap in intensity distribution between the source domain and the target domain.

**PMN [8]** PMN [8] aims to overcome the bottleneck of the learnability in real RAW denoising by reforming the data. It can be generalized and applied to all real RAW image denoising methods and improve their performance including our ADL. As illustrated in Table 3, the performance of our ADL can be improved by applying the training strategy proposed in PMN.

| Method | G4 | GP | IP | N6 | S6 | Avg. |
|---|---|---|---|---|---|---|
| Ours | 52.55/0.975 | 45.18/0.923 | 54.37/0.987 | 46.13/0.932 | 40.16/0.901 | 47.68/0.944 |
| **Ours+PMN** | **52.78/0.976** | **45.32/0.923** | **54.48/0.988** | **46.29/0.933** | **40.30/0.902** | **47.74/0.944** |

| Method | Sony | Fuji | Nikon | Canon | Avg. |
|---|---|---|---|---|---|
| Ours | 37.28/0.871 | 37.58/0.872 | 36.74/0.866 | 36.45/0.868 | 37.01/0.868 |
| **Ours+PMN** | **37.43/0.872** | **37.89/0.874** | **36.91/0.869** | **36.63/0.868** | **37.19/0.871** |

Table 3: **The quantitative PSNR and SSIM results with and without the training strategy proposed in PMN [8] on the SIDD (G4, GP, IP, N6, and S6), ELD (Nikon, and Canon), and SID (Fuji and Sony) datasets.** PMN can improve the performance of our ADL. The camera name on the top row means that we keep this camera as the target domain and use data from other cameras as the source domain.

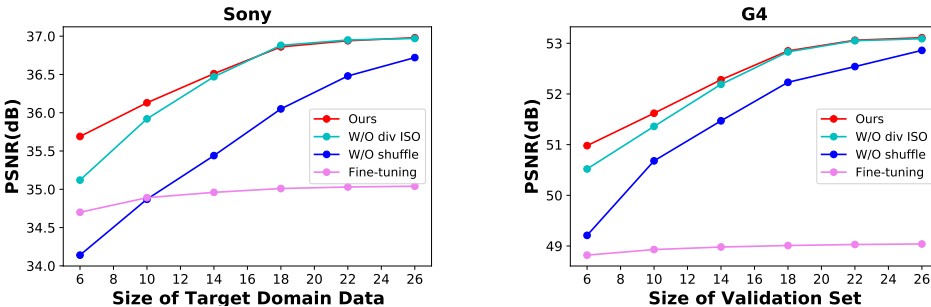

Figure 3: **The ablation study of the size of the validation set.** Our dynamic validation set strategy can overcome the overfitting problem when the size of the target domain dataset is extremely small.

## 4.4 Analysis on Useful and Harmful Data

The domain gap in the deep learning model of other tasks is usually caused by the scene between the training data and test data. As for RAW image denoising, the difference in noise intensity among different sensors is more crucial. For example, if the noise intensity of two sensors is the same, then the noise model trained on one sensor can be generalized to the other. On the contrary, the model will fail if the noise intensity between these two sensors is very different. Based on this observation, we can say that the source domain data with similar noise intensity to the target domain data is useful, while the data with a large noise intensity difference is harmful. However, in cross-domain training, the harmful data has a more negative impact, because in cross-domain training, the model will tend to compromise different domains to reach the global minimum. As illustrated in Table 5, we utilize the target domain data from the ELD dataset as the base set, and we build two Harmful datasets. Here "Harmful1" is synthesized by using the ground truth of the SIDD dataset that is captured in bright light conditions and naive Gaussian noise with noise level "harmful" $\sigma = 30$, and "Harmful2" set is the data pairs that have mis-alignment. For example, the input and ground truth are from different scenes, or the ground truth is black. In these cases, even though we add more data, the performance still drops. However, our ADL ignores the harmful data and always optimizes the model towards the noise intensity of the target domain.

## 4.5 Ablation Study

**ADL and Modulation Strategy** We conduct an ablation study on the effectiveness of our ADL and modulation strategy. The ablation study is conducted on SID and ELD datasets with the same training settings and configuration as in the previous section. We ablate over the strategies of our method by training models without applying the target domain pretraining, source domain adaptive learning, and modulation(including sensor type and ISO modulation). As illustrated in Table 4, the result demonstrated that the target domain pretraining, source domain adaptive learning, ISO and sensor type modulation, and dynamic validation set all contribute to the final result. The source domain adaptive learning, which automatically evaluates whether data from the source domain is harmful or not, is the most crucial strategy in our framework.

| ADL | ISO | Type | Pre | Dyn | Sony | Fuji | Nikon | Canon |
|---|---|---|---|---|---|---|---|---|
| ✓ | ✓ | ✓ |  | ✓ | 36.15/0.858 | 36.44/0.859 | 36.52/0.861 | 36.00/ 0.857 |
| ✓ |  | ✓ | ✓ | ✓ | 37.13/0.868 | 37.41/0.871 | 36.66/0.860 | 36.27/0.857 |
| ✓ | ✓ |  | ✓ | ✓ | 36.81/0.862 | 36.93/0.864 | 36.46/0.858 | 36.11/0.855 |
|  | ✓ | ✓ | ✓ |  | 35.88/0.855 | 36.14/0.856 | 35.97/0.856 | 34.69/0.788 |
| ✓ | ✓ | ✓ | ✓ |  | 36.89/0.866 | 37.41/0.871 | 36.42/0.862 | 36.23/0.861 |
| ✓ | ✓ | ✓ | ✓ | ✓ | **37.28/0.871** | **37.58/0.872** | **36.74/0.866** | **36.45/0.864** |

Table 4: **The PSNR and SSIM result of the ablation study on the ELD and SID datasets.** The camera name on the top row means that we keep this camera as the target domain and use data from other cameras as the source domain. "ADL" means Adaptive domain learning, "ISO" means use ISO in modulation, "Type" means use sensor type in modulation "Pre" means target domain pretraining, and "Dyn" means dynamic validation set.

| Sensor | Base | Base+Harmful1 | | Base+Harmful2 | |
|---|---|---|---|---|---|
| | FT | FT | ADL | FT | ADL |
| Sony | 35.01/0.805 | 34.59/0.772 | 35.13/0.812 | 19.06/0.216 | 34.99/0.808 |
| Fuji | 34.97/0.806 | 34.69/0.771 | 35.21/0.823 | 20.14/0.244 | 35.06/0.807 |
| Nikon | 34.68/0.782 | 34.42/0.765 | 35.85/0.853 | 21.26/0.297 | 34.62/0.782 |
| Canon | 34.76/0.794 | 34.37/0.752 | 34.88/0.797 | 21.17/0.268 | 34.71/0.792 |

Table 5: **The PSNR and SSIM result of our ADL comparing to naive fine-tuning on base set and synthetic harmful dataset.** Here "FT" means naive fine-tuning, "Base" means only using data from the corresponding sensor to fine-tune, while "Harmful1" means using naive Gaussian synthetic data with the different light conditions to fine-tune, and "Harmful2" means using the misaligned input and ground truth data to fine-tune.

**Size of the Target Domain Data** To evaluate how our method performs on different sizes of the target domain data, we conduct the ablation study on two sensors, G4 in the SIDD dataset and Sony in the SID dataset. Figure 3 demonstrates the PSNR against the size of the target domain dataset $T^{adp}$ compared to the fundamental baseline fine-tuning and our method without using the dynamic validation set strategy and diverse system gain selection strategy. It can be observed that our method can outperform fine-tuning when the size of the target domain data is extremely small. Besides, our dynamic validation set strategy also prevents the training from over-fitting when the target domain data is scarce.

## 5    Conclusion

We have proposed a novel adaptive domain learning (ADL) scheme for cross-domain image denoising. We leverage the data from other sensors to help the training of the data from new sensors in a smart fashion: ADL removes harmful data and utilizes useful data from the source domain to improve the performance in the target domain. Our proposed modulation strategy provides extra camera-specific information, which helps differentiate the noise patterns of input data. We evaluate our method on smartphone and DSLR camera datasets, and the results demonstrate that our method outperforms state-of-the-art approaches in cross-domain image denoising. Moreover, we show that ADL can also be easily extended to image deblurring. We believe ADL is general and can be generalized to other cross-domain tasks, which can be further explored in the future.

## Acknowledgement

We thank SenseTime Group Limited for supporting this research project.

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

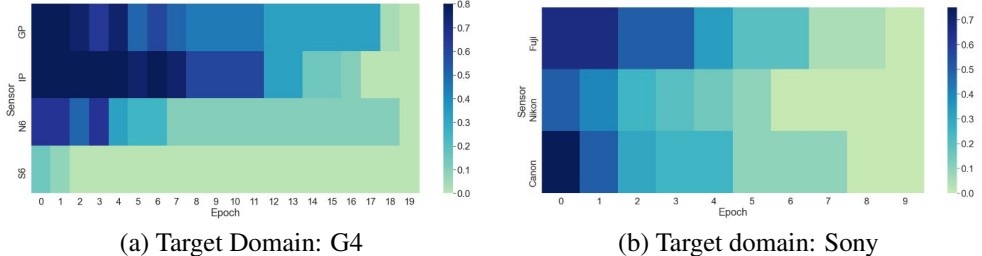

|(a) Target Domain: G4|(b) Target domain: Sony|

Figure 4: **The heatmap of the training process of our model.** The x-axis is the number of epochs, and the y-axis is the sensor. The value in each block ranging from 0-1 indicates the percentage that is judged to have contributed to the training of the target domain. (a) is the training process of sensor G4 in the SIDD dataset. (b) is the training process of sensor Sony in SID dataset.

# A  Appendix

## A.1  Network Architecture

We present the detail of our network architecture. The detail design of our modulation strategy is illustrated in figure 6. We utilize Dense-Unet as our backbone. We downsample the input feature map 4 times and the number of channels is set to 64, 128, 256, 512, and 1024 respectively. There is a total of 5 layers in each dense block. Leaky ReLU [30] is used as the activation function in each layer.

## A.2  Analysis on the Training Process

We investigate how our network uses the data from the source domain to help the training of the target domain. We draw a heatmap to demonstrate the training process of our network. As illustrated in Figure 4, we show the training process of sensor G4 on the SIDD dataset. The x-axis represents the epoch number during the training, and the y-axis represents the type of sensor. Each block in the figure represents the percentage of the data that is judged to have contributed to the training of the target domain in a particular epoch. For example, the value in the top left corner in Figure 4 is 0.75, which means that 75% of the data from sensor GP has contributed to the training of the target domain in the first epoch. It can be seen that at the beginning of the training, most of the data from the source domain contributed to the training of the target domain. As the epoch number and the size of the dynamic validation set increase, more and more data has no contribution to the training of the target domain, which means the useful information hidden in the source domain has been discovered.

Another information we can observe from this heatmap is which sensor from the source domain has a much smaller domain gap compared to the target domain. As illustrated in Figure 4 (a), sensor S6 has only 15% of data that has contributed to the training of the target domain, and the percentage quickly reduces to zero in the following epoch. This indicates that there is a huge domain gap between sensor S6 and the target domain sensor G4, and the data from sensor S6 has almost no contribution to the training of G4. In contrast, the percentage of data usage from sensor IP keep at a high range as the epoch number increases, which indicates that sensor IP has the smallest domain gap to the target domain compared to the other sensors. Moreover, comparing  4 (a) and (b), we can tell that the data captured by the DSLR camera has a larger domain gap than the data captured by the smartphone. This is because the sensor of a DSLR camera is more complicated and has more noise sources than the sensor of a smartphone. Based on the observations, we can sample data with different percentages of the sensors from the source domain and shorten the training time.

## A.3  Extention to Image Restoration Tasks

We demonstrate that our framework can also be applied to image deblurring and image dehazing to overcome the domain gap between the synthetic data and real data. For image deblurring, we synthesize blur kernel with three different sizes $21 \times 21$, $31 \times 31$, $41 \times 41$ by following the blur generation method proposed in  [19]. We perform our experiment by training a Dense-Unet [10] on

| Task | Method | PSNR | SSIM |
|---|---|---|---|
| | Direct | 28.61 | 0.918 |
| Deblur | Fine-tuning | 29.26 | 0.923 |
| | Ours | **30.17** | **0.927** |
| | Direct | 17.34 | 0.618 |
| Dehazing | Fine-tuning | 19.38 | 0.658 |
| | Ours | **19.87** | **0.661** |

Table 6: **Comparison of our method against other normal approaches in image deblurring and image dehazing.** "Direct" means a model trained on synthetic blurry images and then tested on the Gopro dataset. "Fine-tuning" means a model trained on synthetic blurry images, then fine-tuned on a small set of real-world blurry images, and finally testing on the Gopro dataset.

| Dataset | Sensor | Zhang et al. | Single | | Multiple | |
|---|---|---|---|---|---|---|
| | | | FT | ADL | FT | ADL |
| | GP | 0.928 | 0.929 | **0.930** | 0.927 | **0.933** |
| SIDD | S6 | 0.917 | **0.918** | 0.918 | 0.911 | **0.921** |
| | IP | 0.986 | 0.989 | **0.991** | 0.987 | **0.991** |
| ELD | Sony | 0.923 | 0.923 | **0.923** | 0.921 | **0.925** |
| | Nikon | 0.916 | 0.917 | **0.919** | 0.913 | **0.921** |

| Sensor | Led [16] | Ours |
|---|---|---|
| Sony | 0.866 | 0.869 |
| Fuji | 0.867 | 0.872 |
| Nikon | 0.862 | 0.868 |
| Canon | 0.862 | 0.865 |

(a) **The SSIM result of applying our ADL to fine-tune the existing noise calibration model.** Here "FT" means naive fine-tuning, "Single" means only using data from the corresponding sensor to fine-tune, while "Multiple" means using data from all sensors in the corresponding dataset to fine-tune.

(b) **The PSNR result of our ADL comparing to Led on the existing data from various sensors.**

Table 7: The analysis of calibration model and non-calibration model. The SSIM result is included in the supplementary material.

a popular deblurring dataset Gopro [28]. We apply synthetic blur kernels on the ground-truth clear images and form 320 training image pairs as the source domain, and randomly select other 8 training pairs in the Gopro dataset as the target domain training set $T^{train}$ and 16 images to form the target domain validation set $T^{val}$. The other 50 images are used as the test set, with no overlapping on the scene between the validation set, training set, and test set. For image dehazing, we utilize the real-world indoor dehazing dataset, I-haze [3] as the target domain and the synthetic dataset RESIDE [23] as the source domain. We employ the state-of-the-art dehazing method Dehazeformer [36] as the backbone. We evaluate the performance of our result by comparing it with the fine-tuning pipeline using two evaluation metrics, PSNR and SSIM. The quantitative results are illustrated in Table 6. Here, "direct" means training on synthetic data and directly testing on real data without any fine-tuning. The quantitative result demonstrates that our method has higher PSNR and SSIM than fine-tuning.

## A.4 Additional Qualitative Result

We demonstrate additional qualitative results of our method against Blind2unblind [41] and transfer [17] on smartphone dataset SIDD [2] and DLSR datasets ELD [42] and SID [5], as illustrated in Figure 5. We can see that the error between the ground truth and our output noise-free image is much smaller compared to all baseline methods.

We also demonstrate the qualitative result of our method compared to the fundamental baseline fine-tuning on image denoising and image dehazing in figure 7. Our method is able to recover images that are much cleaner and finer.

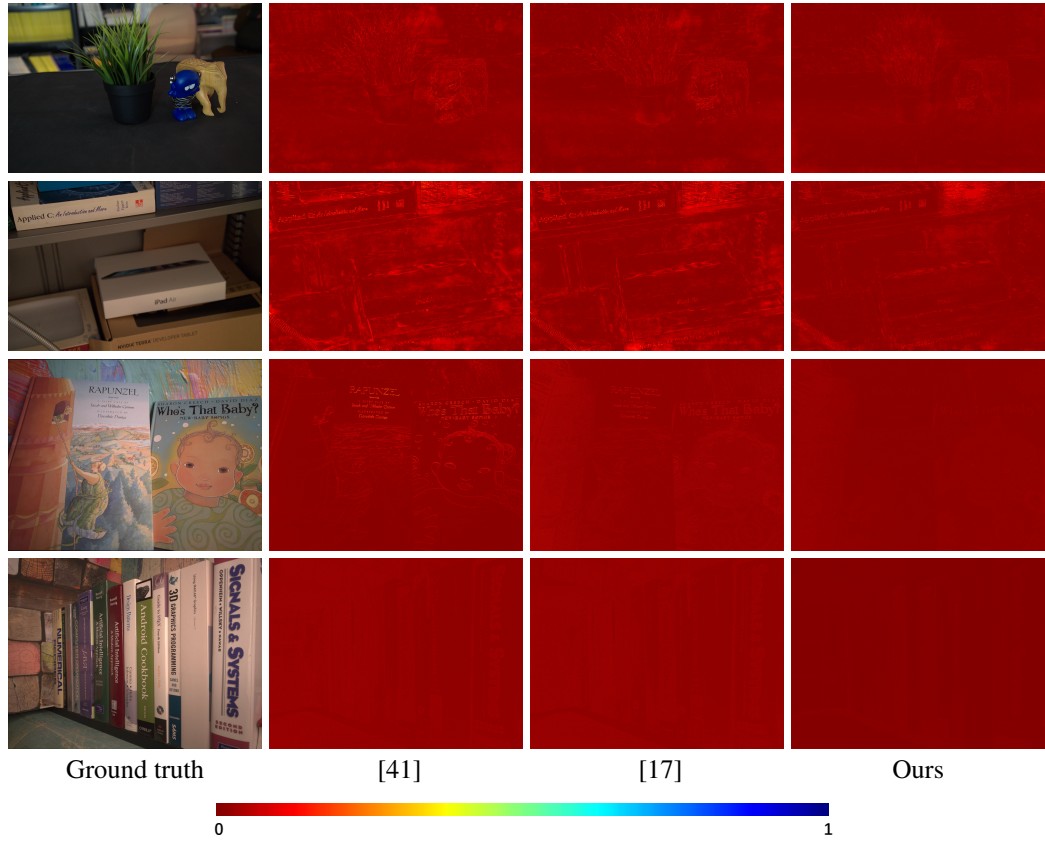

| Ground truth | [41] | [17] | Ours |

0                 1

Figure 5: **The error map of our method compares against state-of-the-art approaches.** The first row is the result from the SIDD dataset, and the second row is the result from the SID dataset. We can see that our method is able to generate the image with smaller errors and less noise compared to previous work.

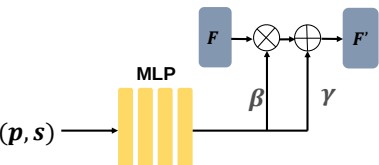

Figure 6: **The illustration of modulation strategy.** The camera-specific metadata $p$ and $s$ (denote the phone code and ISO in our experiments) are transformed into a channel-wise scale $\beta$ and shift $\gamma$ by MLP. Then the convolutional feature $F$ in the network is multiplied by $\beta$ and added by $\gamma$ and obtain $F'$. Note that our modulation strategy is more flexible and can provide more hyper information than the prior work [17, 13, 24, 33].

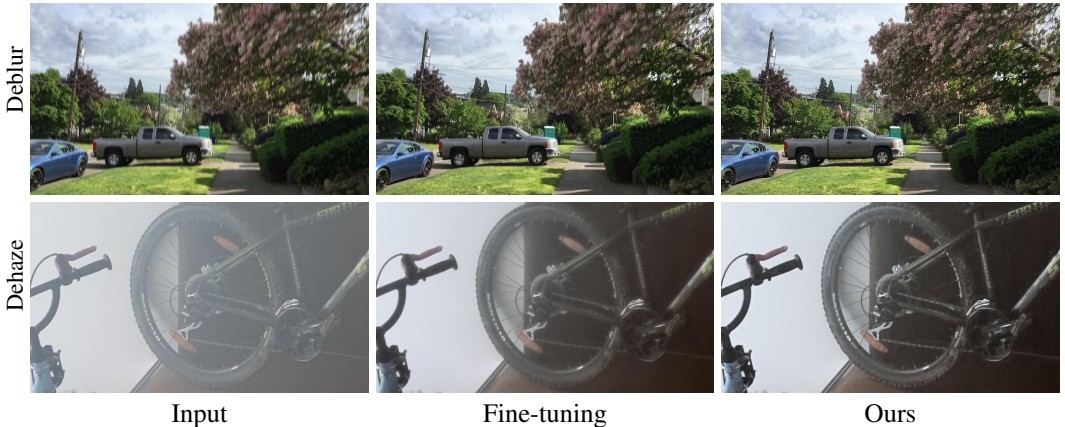

|        | Input | Fine-tuning | Ours |
|--------|-------|-------------|------|

Figure 7: **Comparison between our framework and the fine-tuning pipeline on image deblurring and image dehazing.** Our method is able to generate an image that is much clearer. Zoom in for details.

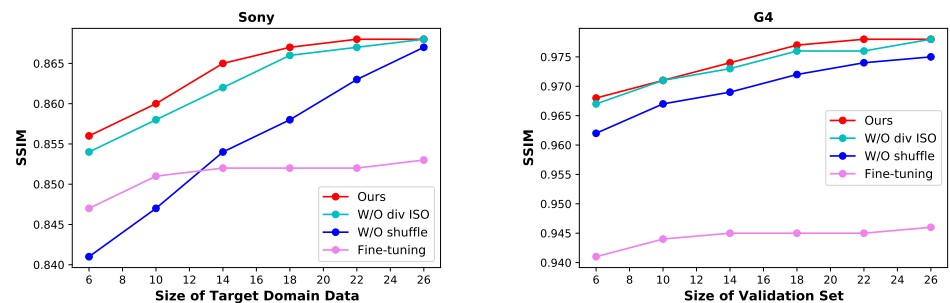

Figure 8: **The ablation study of the size of the validation set.** Our dynamic validation set strategy can overcome the overfitting problem when the size of the target domain dataset is extremely small.

## A.5   Additional Quantitative Result

We demonstrate the SSIM value for section 4.3 and section 4.5. The results are demonstrated in Table 7 and Table 8.

