# OpenReview forum: "Adaptive Domain Learning for Cross-domain Image Denoising"
_NeurIPS.cc/2024/Conference — NeurIPS 2024 poster_

### Official Review · Reviewer_G2YE · 2024-07-08

**Soundness:** 2
**Presentation:** 2
**Contribution:** 2
**Rating:** 4
**Confidence:** 4

**Summary:**

This paper introduce adaptive domain learning (ADL) for cross domain raw image denoising problem. The author also introduce a module to embed sensor information into the network. Experiments on public datasets with various smartphone and DSLR cameras demonstrate proposed model outperforms prior work on cross-domain image denoising with a small amount of image data from target domain.

**Strengths:**

1. The proposed pipeline is straightforward and the idea is easy to understand.
2. It's interesting to apply domain adaptation idea for raw image denoising.

**Weaknesses:**

1. Novelty Issue: Although I am reluctant to mention this, the proposed pipeline is quite common and widely available in domain adaptation research. The main credit is that the author applied it to raw image denoising. However, it may require more innovative insights to meet the standards of the NeurIPS community.
2. Embedding sensor information to help the network distinguish the domain is also a common technique used in many research works. I don't believe it can be considered a significant contribution.
3. In line 210, 20 pairs of data in the target domain is actually not a very small number for a raw image dataset. I am curious about the significance of using these target domain data and how it can contribute to real advancements in raw image denoising applications.
4. The author should fine-tune PMN (the recent SOTA method) with the target domain data for a more thorough comparison.
5. Minor Issues: There is a misalignment in Table 2. It would be better to note SSID, ELD, and SID in Table 1 instead of using camera names only.

PMN (Learnability Enhancement for Low-Light Raw Image Denoising: A Data Perspective)

**Questions:**

Primarily comparing PSNR with self-supervised denoising methods is somewhat unfair because the primary advantage of these methods is not high PSNR. Instead, why not structure the experimental section to compare with state-of-the-art performance methods?

---

> ### Author Rebuttal · Authors · 2024-08-05
>
> We thank the reviewer G2YE for your valuable feedback. We now address your concerns below.
>
> ***- Novelty issue***
>
> In this paper, we attempt to solve the problem of lack of RAW data for new sensors ***in a different way*** compared to prior work. The calibration method needs to collect data pairs to build the calibration model, while the self-supervised method is hard to reach state-of-the-art performance because of its basic assumptions. Our method solves this problem by proposing a cross-domain training strategy, ADL. Our method doesn't need to build any calibration model and can make use of the synthetic data and data from other sensors (source domain), which is usually useless for other methods. With the help of very little amount of data from the new sensor (target domain), our method can train a model that can reach state-of-the-art performance by automatically leveraging useful information and removing useless data.
>
>
> ***- Data requirement***
>
> As illustrated in Figure 3 in the main paper, the performance of our method is saturated when we use 18 pairs of data. However, our method can reach satisfactory results when the number of paired data is 6. Note that 18 pairs of data is still a small amount of data compared to other RAW denoising methods. According to the survey made by [1], for DNN-based supervised methods such as SID [2], it takes about 1800 pairs of data for them to reach state-of-the-art performance. For noise calibration methods [3], they need 150-300 pairs of data to build the calibration model. For the self-supervised denoising method [4], they need 12000 noise images, and they still cannot reach state-of-the-art performance.
>
> ***- Compared with PMN***
>
> PMN is a general method used to overcome the bottleneck of learnability by reforming paired real data according to noise modeling. We do not compare our ADL with PMN because its general strategy can be used to improve the performance of any of the RAW denoising methods, including our ADL. We will add the experiment of how PMN can improve our method in the revision.
>
>
> ***- Minor issues***
>
> We thank you for your suggestion and we will modify them in our revision.
>
> ***- Comparison with self-supervised denoising is not fair***
>
> Both our ADL and self-supervised denoising methods aim to solve the problem of data scarcity in RAW data denoising. Therefore, we take the self-supervised denoising method as a reference. Besides, we also compare our ADL with other state-of-the-art methods that have the same settings, like LED and SFRN, that aim to solve the data scarcity problem, as illustrated in Table 2 in our main paper.
>
> [1] Jin, X., Xiao, J. W., Han, L. H., Guo, C., Zhang, R., Liu, X., & Li, C. (2023). Lighting every darkness in two pairs: A calibration-free pipeline for raw denoising. In Proceedings of the IEEE/CVF International Conference on Computer Vision (pp. 13275-13284).\
> [2] Chen, C., Chen, Q., Xu, J., & Koltun, V. (2018). Learning to see in the dark. In Proceedings of the IEEE conference on computer vision and pattern recognition (pp. 3291-3300).\
> [3] Wei, K., Fu, Y., Yang, J., & Huang, H. (2020). A physics-based noise formation model for extreme low-light raw denoising. In Proceedings of the IEEE/CVF Conference on Computer Vision and Pattern Recognition (pp. 2758-2767).\
> [4] Moran, N., Schmidt, D., Zhong, Y., & Coady, P. (2020). Noisier2noise: Learning to denoise from unpaired noisy data. In Proceedings of the IEEE/CVF Conference on Computer Vision and Pattern Recognition (pp. 12064-12072).

---

> > ### Comment · Reviewer_G2YE · 2024-08-09
> >
> > Hello,
> >
> > Thank you for your response. I would like to reply to the contents one by one and discuss with the author as follow:
> >
> > - ***Novelty issue***
> >
> > As in my initial review, I expressed concerns that the proposed pipeline is quite common in domain adaptation research. While I acknowledge the main credit to the authors for applying these methods to the raw image denoising field, I believe the paper may require more innovative insights to inspire readers, as is expected from publications at NeurIPS. For me, novelty is not a decisive factor for rejection if other aspects are excellent, in other words, I believe there is still much room for improvement to elevate the paper's quality to the acceptance standard.
> >
> >  - ***Data requirement and PMN***
> >
> > As presented in the PMN paper, they obtained significantly better results than SFRN, according to my previous test on PMN's code, they utilized about 160 data pairs. This context led me to comment that "20 pairs of data in the target domain is actually not a very small number for a raw image dataset." However, based on Figure 3 and the author's clarifications, the performance with just 6 paired data is substantially lower than with 26 paired data (around a 3 dB difference), which is significant. I am not yet convinced by the author's rebuttal on this point and would appreciate further explanation.
> >
> >  - ***Comparison with self-supervised denoising is not fair***
> >
> > Self-supervised denoising methods typically use no clean data (no supervision), and unsupervised denoising uses unpaired data. In contrast, the method proposed by the authors not only requires paired data from the source domain for supervision but also from the target domain, which should naturally result in better PSNR/SSIM scores. This is why I question the fairness of using these as comparable baselines for performance comparisons. Additionally, the results presented in the tables do not show a significant performance advantage over self-supervised or unsupervised methods, despite with the support of substantial data and supervision signals.
> >
> > - ***Additionally***
> >
> > I am curious about the significance of using these target domain data and how it can contribute to real advancements in raw image denoising applications. Does this domain adaptation strategy have any practical applications in real-world scenarios for raw image denoising?
> >
> > Looking forward to discussion with you.
> >
> > Best regards,
> >
> > Reviewer G2YE

---

> > > ### Author Response · Authors · 2024-08-13
> > >
> > > We thank you for affirming and approving the application of ADL to RAW denoising and other clinical aspects and for your valuable suggestions. We will improve our paper further in the revision. Besides, we would really appreciate it if you could point out the specific work that a similar pipeline is proposed in domain adaptation research so that we can learn from it and include it in our literature review. We were eager to address any further questions or concerns you may have. If you have had a chance to review our response and have additional thoughts, we would greatly appreciate your input.

---

> ### Author Response · Authors · 2024-08-10
>
> Thank you for your reply. We would like to further discuss the question with you as below:
>
> ***- Data requirement and PMN***
>
> Collecting paired real data for RAW denoising requires considerable human labor and equipment support. Collecting 160 pairs of real data requires a lot more effort than collecting 20 pairs of real data. The PMN can reach better performance with less data because they overcome the bottleneck of learnability in real RAW denoising. Moreover, PMN is a data reformation method, their method can be applied to any training strategy and network architecture and has similar performance improvement. We applied the DSC strategy proposed in PMN to our ADL. The PSNR on the Sony sensor improved by 0.41dB, and the PSNR on the Nikon sensor improved by 0.38 dB.
>
> ***- Comparison with self-supervised denoising is not fair***
>
> We try our best to find all possible work that has the ***same goal*** as our method for the ***comprehensive*** of the baselines.  We compare our ADL with the self-supervised denoising method just for reference. Moreover, as illustrated in section 4.1, line 211 in the main paper, in our experiment,  the self-supervised denoising method did not use any source domain data and used all the data from the target domain data, which is the same as their own setting. Compared to our ADL, they use a lot more data and do not involve any cross-domain learning.
>
> ***- The practical applications of ADL***
>
> Here are two examples of practical applications for our ADL in real-world scenarios:
> 1. As the iteration of the smartphone and DLSR cameras become faster and faster in recent years, collecting a large RAW denoising dataset for each of these sensors to build noise calibration models or single domain supervised learning is very labor-demanding. Moreover, the dataset for specific sensors cannot be used in the training of sensors in the future and therefore causes a waste of resources. With the help of our ADL, we only have to collect a small dataset with around 20 pairs of data. Besides, the dataset we collect for old sensors can also be reused and serve as the source domain to help with the training of the new sensors.
> 2. Collecting paired real RAW images is difficult and may collect bad data, such as misalignment, without the help of professional equipment, such as robot arms. Moreover, synthetic data may also have outliers, which are very different from the noise distributions of real-world data. These bad data are hard to detect and can lead to significant performance drops in supervised learning. As illustrated in section 4.4, the robustness of our ADL can avoid the performance drop brought by these bad data.

---

> > ### Comment · Reviewer_G2YE · 2024-08-14
> >
> > Hello,
> >
> > Thank you for your response.
> >
> > After careful consideration of all the rebuttals and available reviews/discussions, I have decided to maintain my original score. I believe this to be the most appropriate reflection of my overall impression and evaluation of the paper.
> >
> > Best regards,
> >
> > Reviewer G2YE

---

### Official Review · Reviewer_Xk6Q · 2024-07-11

**Soundness:** 4
**Presentation:** 3
**Contribution:** 3
**Rating:** 6
**Confidence:** 5

**Summary:**

This paper proposes a novel adaptive domain learning (ADL) approach for cross-domain RAW image denoising problem. The ADL scheme allows to train models for target domains with limited data by leveraging data from other source domains. The harmful data from the source domain is automatically removing during training. Authors also propose a new modulation approach to encode the sensor type and ISO to help the network to adapt to different noise distributions. The extensive experimental results demonstrate the effectiveness of the proposed method.

**Strengths:**

1. The proposed ADL training strategy is useful in the cases of very limited target domain data and is able to effectively utilize the existing source domains by filtering out harmful data
2. Extensive ablation studies demonstrate the robustness of the method even in the presence of misaligned data
3. The experiment results in the supplementary show that the method generalizes to other problems (dehazing, deblurring)

**Weaknesses:**

1. The qualitative results only show the error maps. Actual denoised images (or patches) in the sRGB should be provided for better visual comparison of the methods. In addition, quantitative analysis with metrics such as LPIPS should be done on the sRGB methods
2. It seems that the already trained network cannot easily adapted to new data. The method encodes the sensor type as a one-hot vector. Therefore, the new source domain data (new sensor) cannot be incorporated to fine-tune the network and the training should be done from scratch.

**Questions:**

Have you tried other metrics than PSNR for dynamic validation?

---

> ### Author Rebuttal · Authors · 2024-08-05
>
> We thank the reviewer Xk6Q for your valuable feedback. We now address your concerns below.
>
> ***- Qualitative results and metrics in sRGB space should be provided for better comparison***
>
> Since RAW data is hard to visualize and tell the difference, we use the error map for better demonstration. We will put some comparisons in sRGB space in our revision.
>
> ***- It is hard to incorporate new source domain data because of the one-hot encoding***
>
> It is not hard to incorporate new source domain data into the model. To maintain the network's ductility, you can keep the size of the one-hot encoding vector greater than the number of types of sensors in the source domain data. For example, if you have 5 sensors as the source domain data, you can make the one-hot encoding vector size 6 to add potential source domain data in the future, and this will not affect the training for the current stage.
>
> ***- Other metrics than PSNR for dynamic validation***
>
> As illustrated in Figure 1 in the rebuttal pdf file, we demonstrate the dynamic validation set ablation study on the SSIM metric.

---

> > ### Comment · Reviewer_Xk6Q · 2024-08-09
> >
> > I appreciate the authors' efforts in addressing my comments and providing additional experimental results for the SSIM metric. However, I am still not convinced that the method is flexible for fine-tuning new source domain data. Even though the one-hot encoding vector size can be set higher than the number of sensor types, we can still have more new sensor types in the future that exceed the vector size

---

> > > ### Author Response · Authors · 2024-08-10
> > >
> > > We thank you for you valuable feedback. We would like to further discuss the problem with you.
> > > You can set the one-hot encoding vector size to be some large number, for example, greater than 100. We believe that 100 is enough for real practice. Very large number of source domains barely appear in real scenarios.

---

> > > ### Author Response · Authors · 2024-08-13
> > >
> > > We value your feedback and are eager to address any further questions or concerns you may have. If you have had a chance to review our response and have additional thoughts, we would greatly appreciate your input.

---

### Official Review · Reviewer_e1pc · 2024-07-12

**Soundness:** 3
**Presentation:** 2
**Contribution:** 2
**Rating:** 5
**Confidence:** 4

**Summary:**

This paper address the cross-domain image denoising problem with a small number of target domain training samples. The authors propose an adaptive domain learning (ADL) strategy that dynamically selects useful training samples from both source and target domains to improve performance.  Additionally, the paper leverages channel-wise modulation layer to model different noise distribution caused by sensor type and ISO. Experimental results showcase the advantages of the proposed method.

**Strengths:**

1. The paper clearly defines the problem of few-shot domain adaptation for image denoising and presents a solution.

2. The reported quantitative results in the paper indicate improved performance over existing methods, suggesting potential consideration and applicability in real-world scenarios.

3. In general the writing is clear and easy to follow, with minor presentation issues.

**Weaknesses:**

1. **Lacking Clarity on Definitions and Practical Benefits:**  The paper actually discusses the few-shot domain adaptation problem but fails to clearly define the specific settings and scenarios being addressed. Also, this paper does not clearly specify the conditions under which the proposed ADL strategy provides the most benefit. For example, how does the performance of ADL compare with simpler strategies like mixing the source and target domain data as the number of target samples increases? What is the threshold number of samples where ADL proves advantageous? Is this number large enough so that collecting target domain data larger than the threshold can be seen as impractical.

2. **Lack of comparison to adequate baselines**: The paper would benefit from a more comprehensive comparison with existing  few-shot domain adaptation method in low level vision [1,2]. Comparing the proposed method with approaches in different settings, such as self-supervised learning, may not be fair or relevant. It would be more informative and meaningful to compare against methods with similar objectives and constraints.

3. **Lack of Task-Specific Design and Motivation**: The proposed ADL strategy is not specifically tailored for the denoising task, which reduces its effectiveness and relevance. Additionally, the paper does not clearly explain the motivation behind combining ADL with the modulation module. The paper would benefit from a clearer explanation of how these components interact and complement each other.

4. **Presentation Issues:** There are inconsistencies in the presentation of results, such as the caption in Table 3 mentioning "Mod" (modulation) without corresponding columns in the table.

[1] K. Prabhakar, Vishal Vinod, et al. "Few-Shot Domain Adaptation for Low Light RAW Image Enhancement." British Machine Vision Conference 2021.

[2] Bo Jiang, Yao Lu, et al. "Few-Shot Learning for Image Denoising." 2023.

**Questions:**

1. As the size of the target domain dataset increases, how does the performance of the ADL strategy compare with simpler strategies such as mixing the source and target domain datasets? Is there a threshold number of target samples beyond which ADL does not offer significant benefits?

2. What are the computational overheads associated with the dynamic validation set used in the ADL strategy? Is the process efficient enough?

**Limitations:**

Limitations are currently not discussed in this paper, which may not be so adequate.

---

> ### Author Rebuttal · Authors · 2024-08-05
>
> We thank the reviewer e1pc for your valuable feedback. We now address your concerns below.
>
> ***- Lacking Clarity on Definitions and Practical Benefits***
>
> Our ADL aims to solve the problem of data scarcity in RAW image denoising. To be specific, our ADL has benefits when the data pairs from the target domain are limited. Our ADL can utilize data from the source domain, including synthetic data and real data from other sensors to improve the performance of the training using target domain data. As illustrated in Figure 3 in the main paper, we compare our ADL with naive fine-tuning (pre-trained on source domain data and fine-tuning using target domain data) as the size of target domain data increases. Our ADL outperforms naive fine-tuning when the target domain data is scarce. Besides, according to our experiments, naive fine-tuning can have comparable results with our ADL when the size of target domain data is around 280 on the sensor Sony.
>
> ***- Lack of comparison to adequate baselines***
>
> We compare our ADL with [1], as illustrated in Table 1 in the rebuttal pdf file. Our method can outperform their method on PSNR and SSIM. We do not compare with [2] because they did not open source. We will include them in the literature review in our revision.
>
> ***- Lack of Task-Specific Design and Motivation***
>
> We combined the modulation strategy with ADL to adjust the feature space by embedding two easy-to-access parameters, the sensor type and the ISO. These two sensor-specific information help our network get the knowledge of two crucial aspects in cross-domain RAW image denoising, the domain gap and the noise level. Each of the sensors has a different domain gap compared to the target domain. Taking the sensor type as input, our network can explicitly judge the domain gap between the input source domain data and the target domain and further adjust the features. ISO is the sensor's key setting that affects the noise level of the captured RAW images. Taking ISO as the input, our network can get the knowledge of how noisy the input RAW images are and therefore can adjust the features to fit different noise levels better.
>
> ***- Presentation Issues***
>
> We thank you for your suggestion and we will modify them in our revision.
>
> ***- Computational overheads of dynamic validation set***
>
> The dynamic validation set strategy does not bring much computational cost to our framework. On RTX4090 GPUs, dynamic validation takes 0.21s per image.
>
> [1] K. Prabhakar, Vishal Vinod, et al. "Few-Shot Domain Adaptation for Low Light RAW Image Enhancement." British Machine Vision Conference 2021. \
> [2] Bo Jiang, Yao Lu, et al. "Few-Shot Learning for Image Denoising." 2023.

---

> > ### Comment · Reviewer_e1pc · 2024-08-11
> >
> > Thanks to the authors for their rebuttal and extra experiment. However, some of my questions have not been properly answered:
> >
> > **Q1**: Regarding the setting, this approach requires source data in domain adaptation. However, source data is not available in many cases (source-free domain adaptation). If those compared methods are source-free, the comparison may not be fair.
> >
> > **Q2**: We are happy to see the additional results. However, the authors did not address my concern: "Comparing the proposed method with self-supervised learning may not be fair," because SSL does not require target domain data with ground truth.

---

> > > ### Author Response · Authors · 2024-08-12
> > >
> > > Thank you for your reply. We would like to further discuss the question with you as below:
> > >
> > > ***- Unfair comparison with the baselines***
> > >
> > > We try our best to find all possible work that has the ***same goal*** as our method for the ***comprehensive*** of the baselines. We compare our ADL with the source-free domain adaptation and self-supervised denoising method just for reference, and we will emphasize this point in our revision. The source domain data of our ADL can be easily accessed. The source domain data could be synthetic data or data from any other sensors. Due to the robustness of our ADL, the harmful data from the source domain will be automatically removed. For the self-supervised denoising method, as illustrated in section 4.1, line 211 in the main paper, in our experiment, the self-supervised denoising method did not use any source domain data and used all the data from the target domain data, which is the same as their own setting.

---

> > > > ### Comment · Reviewer_e1pc · 2024-08-14
> > > >
> > > > Thank you for your clarification. I am willing to raise my score.

---

> > > > > ### Author Response · Authors · 2024-08-14
> > > > >
> > > > > Thank you very much for the score improvement and your constructive feedback. We will further polish the paper in the final revision.

---

> > > ### Author Response · Authors · 2024-08-13
> > >
> > > We value your feedback and are eager to address any further questions or concerns you may have. If you have had a chance to review our response and have additional thoughts, we would greatly appreciate your input.

---

### Official Review · Reviewer_4VKx · 2024-07-12

**Soundness:** 3
**Presentation:** 3
**Contribution:** 2
**Rating:** 5
**Confidence:** 4

**Summary:**

This paper addresses the challenge of cross-domain RAW image denoising due to varying noise patterns from different camera sensors (bit depths, color). The authors propose an Adaptive Domain Learning (ADL) scheme that leverages existing data from various sensors and a small amount of data from a new sensor to train an effective denoising model. The ADL scheme selectively uses beneficial data from the source domain while discarding harmful data, and introduces a modulation module to incorporate sensor-specific information. The proposed model outperforms prior methods on public datasets, demonstrating state-of-the-art performance with limited target domain data.

**Strengths:**

1. Originality: The adaptive domain learning (ADL) strategy is a novel approach to addressing the cross-domain image denoising challenge.
2. Quality: The proposed method is thoroughly evaluated through extensive experiments on various public datasets, showing its robustness and effectiveness.
3. Clarity: The paper is well-structured, with clear explanations of the methodology, experiments, and results.

**Weaknesses:**

1. Compared with RAW2RAW mapping: There are existing methods for raw-to-raw mapping, such as "Semi-Supervised Raw-to-Raw Mapping." It would be interesting to explore whether source domain raw images can be mapped to the target domain and then used for training based on the noise model. This could potentially provide a more direct comparison and integration of data from different domains.

2. Data Requirements: The authors used over 20 captured raw images, which is significantly more than the "Two pairs" approach used in LED. With these raw images, could a noise model be used to simulate noise instead? Is there a comparison available? This would help to understand whether the additional raw images provide a significant advantage over using simulated noise from a noise model.

**Questions:**

The current ADL settings require more data compared to the LED method. In Table 2b, it is not clear how the LED method was specifically trained—whether it involved fine-tuning with limited target domain data. Additionally, comparing the performance of ADL with LED under the "Two pairs" condition would provide valuable insights into the effectiveness and efficiency of both approaches. This could help understand the trade-offs between data requirements and performance outcomes.

**Limitations:**

A more explicit discussion on the potential impact of extremely limited target domain data (like two pairs) on the model's performance would provide additional clarity.

---

> ### Author Rebuttal · Authors · 2024-08-05
>
> We thank the reviewer 4VKx for your valuable feedback. We now address your concerns below.
>
> ***- Compared with RAW2RAW mapping***
>
> We mapped source domain RAW data to target domain data by the pre-trained model proposed in [1]. We utilize sensors "IP" and "S6" from SIDD dataset and replace the corresponding metadata in the data preprocessing stage of [1] to keep the data consistent. The following table demonstrates our ADL compared to RAW-to-RAW mapping by [1]. Here "IP" means mapping sensor S6 to sensor IP and "S6" means mapping sensor IP to sensor S6. The value in the table is PSNR/SSIM. The result demonstrates that our ADL outperforms RAW-to-RAW mapping. This is because RAW-to-RAW mapping is an ill-posed problem and hard to handle the overexposure and underexposure cases.
>
> | | IP | S6 |
> | --- | --- |--- |
> |[1]|51.26/0.971|38.17/0.889|
> |Ours|53.09/0.978|39.68 /0.897|
> ***- Data Requirements***
>
> As illustrated in Figure 3 in the main paper, the performance of our method is saturated when we use 18 pairs of data. However, our method can reach satisfactory results when the number of paired data is 6. Note that 18 pairs of data is still a small amount of data compared to other RAW denoising methods and is not enough to build a noise model. According to the survey made by [2], for DNN-based supervised methods such as SID [3], it takes about 1800 pairs of data for them to reach state-of-the-art performance. For noise calibration methods such as [4], they need 150-300 pairs of data to build the calibration model. For the self-supervised denoising method such as [5], they need 12000 noise images, and they still cannot reach state-of-the-art performance.
>
> ***- How is LED trained and how is ADL's performance under the setting of 2 pairs?***
>
> LED is pre-trained on data synthesized by the calibration model built by the simulation camera. In the pre-train stage of LED in Table 2 in our main paper, we replace the synthetic data with our source domain and target domain data to keep the comparison fair. In the fine-tuning stage, LED and ADL use the same target domain dataset with a size of 18. Note that LED actually uses 6 pairs of data (2 pairs for each ratio) in the fine-tuning stage, as illustrated in section 3.3 of LED's main paper. Therefore, we show the performance of our ADL compared to LED using 6 pairs of target domain data. On the Sony sensor, the PSNR of our ADL is 35.82, while the LED is 35.79. On the Nikon sensor, the PSNR of our ADL is 35.56, while the LED is 35.47.
>
> [1] Afifi, M., & Abuolaim, A. (2021). Semi-supervised raw-to-raw mapping. British Machine Vision Conference (BMVC).\
> [2] Jin, X., Xiao, J. W., Han, L. H., Guo, C., Zhang, R., Liu, X., & Li, C. (2023). Lighting every darkness in two pairs: A calibration-free pipeline for raw denoising. In Proceedings of the IEEE/CVF International Conference on Computer Vision (pp. 13275-13284).\
> [3] Chen, C., Chen, Q., Xu, J., & Koltun, V. (2018). Learning to see in the dark. In Proceedings of the IEEE conference on computer vision and pattern recognition (pp. 3291-3300).\
> [4] Wei, K., Fu, Y., Yang, J., & Huang, H. (2020). A physics-based noise formation model for extreme low-light raw denoising. In Proceedings of the IEEE/CVF Conference on Computer Vision and Pattern Recognition (pp. 2758-2767).\
> [5] Moran, N., Schmidt, D., Zhong, Y., & Coady, P. (2020). Noisier2noise: Learning to denoise from unpaired noisy data. In Proceedings of the IEEE/CVF Conference on Computer Vision and Pattern Recognition (pp. 12064-12072).

---

> ### Author Response · Authors · 2024-08-13
>
> We value your feedback and are eager to address any further questions or concerns you may have. If you have had a chance to review our response and have additional thoughts, we would greatly appreciate your input.

---

### Author Rebuttal · Authors · 2024-08-05

We thank the reviewers for their valuable feedback. Below, we address the concerns of Reviewer 4VKx, Reviewer e1pc, Reviewer Xk6Q and Reviewer G2YE.

Our paper presents a novel adaptive domain learning (ADL) method with a modulation strategy to solve the problem of data scarcity in RAW denoising. Our ADL can utilize data from various sensors (source domain) to help the training of very limited data from a new sensor (target domain). Our ADL can automatically remove harmful data from the source domain during the training. As highlighted by the reviewers: "The ADL strategy is novel (Reviewer 4VKx)",  "The experiment shows ADL's robustness and effectiveness (Reviewer 4VKx, Reviewer Xk6Q) and indicates improved performance over existing methods(Reviewer e1pc)",  "The paper is well-structured and easy to follow (Reviewer 4VKx, Reviewer e1pc, Reviewer G2YE)".

---

### Decision · Program_Chairs · 2024-09-25

**Decision:**

Accept (poster)

**Comment:**

This paper proposes a novel Adaptive Domain Learning (ADL) method for cross-domain RAW image denoising, leveraging both source and limited target domain data to enhance performance. While the method demonstrates superior performance across several public datasets and is technically sound, there are concerns regarding its novelty and comprehensive comparison with existing methods. The paper received mixed reviews regarding its contribution and presentation. The authors provided a thorough response addressing the concerns raised by the reviewers, leading to a final average score of 5.25. The AC carefully reviewed the paper and all the reviewers' comments. Considering the good performance of the proposed framework on benchmark datasets and its potential impact for the community, the AC agrees to accept the paper and strongly recommends incorporating the content from the rebuttal into the final version.